# Self/other distinction in adolescents with autism spectrum disorder (ASD) assessed with a double mirror paradigm

Nathalie Lavenne-Collot[1,2,3]*, Marie Tersiguel[1,2], Nolwenn Dissaux[1,2], Céline Degrez[1], Guillaume Bronsard[1,2,4], Michel Botbol[2,5], Alain Berthoz[6,7]

1 Service de Psychiatrie de L'Enfant et de l 'Adolescent, CHRU Brest, Brest, France, 2 Université de Bretagne Occidentale, Brest, France, 3 Laboratoire du traitement de l'information médicale, Inserm U1101, Brest, France, 4 Département de Sciences Humaines et Sociales, EA 7479, EA 3279 (CEReSS, AMU), Brest, France, 5 Professeur Emérite de Psychiatrie de l'Enfant et de l'Adolescent, Université de Bretagne Occidentale, Brest, France, 6 Professeur Honoraire au Collège de France, Paris, France, 7 Centre Interdisciplinaire de Biologie (CIRB), Paris, France

* nathalie.lavenne-collot@chu-brest.fr

## Abstract

### Background

Self/other distinction (SOD), which refers to the ability to distinguish one's own body, actions, and mental representations from those of others, is an essential skill for effective social interaction. A large body of clinical evidence suggests that disruptions in SOD may be key to social communication deficits in individuals with autism spectrum disorders (ASD). In particular, egocentric biases have been found in cognitive, affective, behavioural, and motor domains. However, research in this area is scarce and consists of recognition paradigms that have used only static images; these methods may be insufficient for assessing SOD, given the increasing role of embodiment in our understanding of the pathophysiology of ASD.

### Method

A single-centre, prospective pilot study was carried out to investigate, for the first time, self-recognition and SOD in seven adolescents with ASD compared with matched, typically developing controls (TDCs) using the "Alter Ego"[TM] double mirror paradigm. The participants viewed a double mirror in which their own face was gradually morphed into the face of an unfamiliar other (self-to-other sequence) or vice versa (other-to-self sequence); participants were instructed to indicate at which point the morph looked more like their own face than the other's face. Two judgement criteria were used: 1) M1: the threshold at which subjects started to recognize their own face during the other-to-self morphing sequence; 2) M2: the threshold at which subjects started to recognize the other's face during the self-to-other morphing sequence.

### Results

Consistent with the predictions, the results showed that the participants with ASD exhibited earlier self-recognition in the *other-to-self* sequence and delayed other-recognition in the

**Data Availability Statement:** All relevant data are within the manuscript and its Supporting Information files.

**Funding:** the authors received no specific funding for this work.

**Competing interests:** The authors have declared that no competing interests exist.

*self-to-other* sequence, suggesting an egocentric bias. SOD impairments were also marginally correlated with ASD severity, indicating earlier face recognition in more severely affected individuals. Furthermore, in contrast with that of TDCs, the critical threshold for switching between self and other varied with the direction of morphing in ASD participants. Finally, these differences in face recognition and SOD using mirrors, unlike previous research using static images, support the central place of bodily self-consciousness in SOD impairments.

## Conclusions

Although additional research is needed to replicate the results of this preliminary study, it revealed the first behavioural evidence of altered SOD in ASD individuals on an embodied, semiecological face-recognition paradigm. Implications for understanding ASD are discussed from a developmental perspective, and new research and therapeutic perspectives are presented.

## Introduction

Distinguishing self from others is a fundamental key aspect of social behaviour. The development of this skill entails recognizing when self and other perspectives or experiences are shared and congruent and under which circumstances they differ from one another [1, 2]. Such social experiences involve both an ability to identify with others and an ability to distinguish ourselves from others [3, 4]. In the absence of this capacity for self/other distinction (SOD), confusion between self and other can occur: the experience of others can be confused as coming from the self (i.e., "altercentric" bias), or one can assume an understanding of the other's mind based on one's own experience (i.e., "egocentric" bias) [5–7].

Autism spectrum disorders (ASD) are a lifelong developmental condition involving impairments in social functioning, language or communication and the presence of stereotyped repetitive behaviours and highly restricted interests [8]. In ASD, clinical and neuropsychological observations have highlighted several examples of egocentric bias in cognitive, affective, behavioural, and motor domains, supporting the hypothesis that SOD impairments may play a crucial role in social communication deficits [9–11]. The term "autism" comes from the Greek word "autos" meaning "self", and in his first descriptions, Kanner highlighted egocentricity in self/other processing with numerous references such as "self-absorbed" and "self-satisfied" [12]. In the cognitive domain, ASD patients are assumed to have difficulty taking the perspective of others, and there are notable impairments in theory of mind (TOM) abilities with egocentric bias [13, 14]. In the affective domain, individuals with ASD have been found to experience higher emotional contagion, in that they tend to conflate others' feelings with their own [15–17]. At the behavioural level, many examples of egocentricity can be found, e.g., the use of another's hand, echolalia (involuntary repetition of another person's vocalizations), echopraxia (involuntary repetition of another person's actions), and abnormalities in person deixis and pronoun reversal ("you" instead of "I") [18].

Moreover, successful social interactions rely on the ability to flexibly switch between representations of self and others and inhibit the representation that is not relevant in a given situation [19]. Several authors have suggested that this mechanism, referred to as "self–other control" (SOC), could be a candidate to explain interaction impairments and may be significantly altered in ASD individuals [20, 21].

To date, studies that have explored SOD generally used prerecorded static images or movies that progressively morph from one's own face to another's face and vice versa, both in healthy individuals [22–24] and in patients with psychiatric disorders [25–27]. In particular, this kind of task was performed by Uddin et al. [27] in ASD participants and controls. This study showed no significant differences between the two groups in behavioural performance, percentage of self-reports or reaction time, which might indicate a successful SOD.

However, it can be questioned whether static image recognition paradigms are sufficient for SOD assessment. Indeed, the sense of one's own body, variously termed "embodiment" [28] or bodily self-consciousness [29], is considered, notably in the phenomenological tradition, the cornerstone of mental life and a starting point for theories of the self [30, 31]. In particular, several studies have supported the notion that embodiment is increasingly relevant for understanding clinical conditions, such as autism and psychosis [32, 33], and alterations in embodiment may contribute to a broad range of symptoms and differences in ASD [34]. Therefore, studies of ASD from an embodied perspective are urgently needed, as well as paradigms involving participants who are physically present to improve our understanding of ASD physiopathology [35]. These considerations have led to the search for new paradigms that do not rely on static images but explore SOD in a more ecologically relevant manner.

Exploring SOD using a mirror seems to be another promising avenue. A mirror is indeed a familiar and ecologically relevant everyday tool that plays a crucial role in several theoretical systems, from both developmental and psychodynamic perspectives. An ability for self-recognition in a mirror is indicative of an underlying self-concept and an important behavioural marker of higher-order consciousness [36, 37]. In particular, the ability to differentiate one's own image from that of another in the specular image is considered to be a precursor of self-awareness reached at approximately 4 months old [38]. Moreover, Zazzo [39] described the way in which recognition of others (acquired at 8 months) far precedes self-recognition (acquired at approximately 2 years old) with the progressive awareness of one's own body image concurrent with language development. This suggests that language development, in terms of its social communication aspects, requires self/other differentiation.

Moreover, the use of mirrors and other kinds of image self-processing, in particular, static images, cannot be considered equivalent measures. Previous studies indeed found that (i) mirror self-recognition emerges prior to photo self-recognition [40]; (ii) different neural responses have been found when comparing mirror and photo self-processing [41]; and (iii) preserved self-face recognition on photographs, despite a lack of self-face recognition in mirrors, has been described in some neurological patients [42]. For all these reasons, generalization of findings from SOD tasks using static images to the use of mirrors is not warranted, and there is nothing known about SOD using mirrors in ASD.

Here, for the first time, we examined self-recognition in a mirror and SOD abilities in individuals with ASD using a self-versus-other face identification task through a novel paradigm using the Double Mirror "Alter Ego"™ invented by the artist Moritz Werhmann. This new paradigm developed by Alain Berthoz and Berangere Thirioux allows us to specifically explore SOD under greater ecologically relevant conditions than the use of static images by merging the faces of two subjects physically facing each other and interacting on two sides of the mirror. It has been successfully used in several studies on self/other interactions in healthy subjects [43] and patients with psychiatric disorders, especially schizophrenia [44].

As shown in Fig 1, participants watch a double mirror in which a picture of their own face gradually transforms into the face of an unfamiliar other (*self-to-other* sequence) or vice versa (*other-to-self* direction) and indicate at which point they judge the morph to look more like their own face than the other person's face.

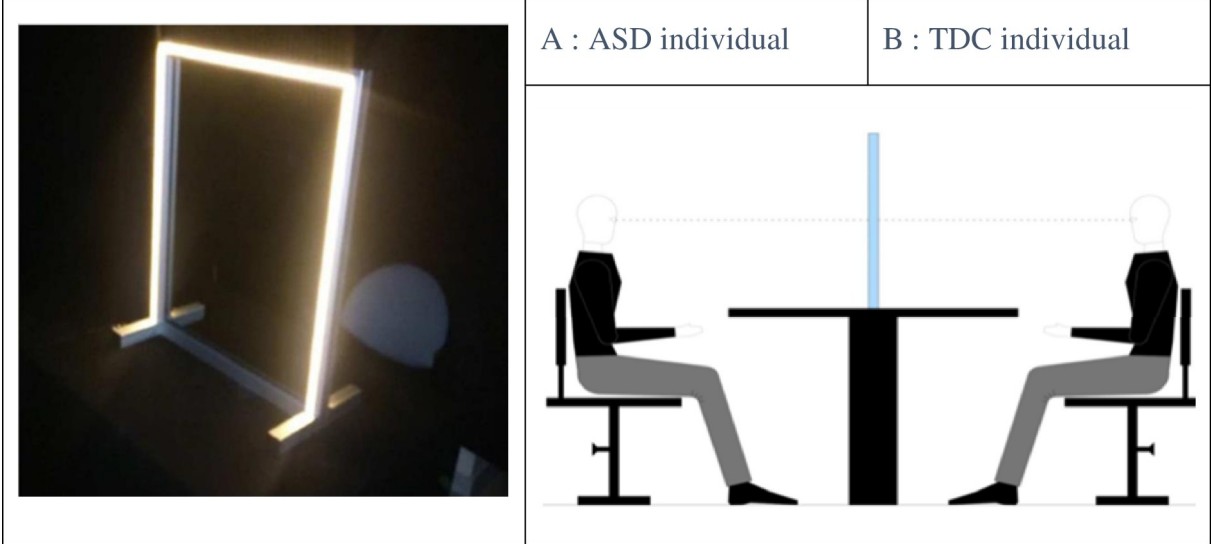

**Fig 1. Experimental setting.** a. Picture of the double mirror Alter Ego designed by Moritz Wehrmann. The experiment took place in a completely darkened enclosed space in the research centre of the Brest Hospital. The mirror was installed on a square table. The ASD patient and his matched control sat facing each other on either side of the double mirror. b. Schematic representation of the individuals during the recognition task. Both participants wore a black cape covering the neck, the whole chest and the arms so that only the face was visible. They were asked to look each other straight in the eye and to focus only on the other person's face. Height-adjustable chairs allowed strict alignment of the eyes.

The objective of the present study was to examine self-recognition and SOD abilities in adolescents with ASD compared to typically developing controls (TDCs) by using the ability provided by this device to progressively change the identity of self and other in the mirror with changes in lighting on both sides of the device. This is a special kind of morphing using real actors, in contrast with the computerized methods usually used in such studies.

In the present study, we had the following hypotheses:

1. First, ASD individuals would show differences in SOD abilities compared with TDCs during the self–other facial double mirror morphing task;

2. Specifically, we hypothesized that the ASD group would show an "egocentric bias" regardless of whether the participants had to switch from *self to other* or from *other to self*;

3. Consistent with this egocentric bias, during the morphing sequence from *self to other*, we expected the ASD group to have more difficulty recognizing the "other" than TDCs.

4. Conversely, during the morphing sequence from *other to self*, we expected the ASD group to have less difficulty recognizing the "self" than TDCs;

5. Finally, we expected that core symptoms of autism may result from this weaker SOD and assessed this possibility by evaluating the relationship between performance in the double mirror SOD task and autism severity.

## Materials and methods

### Participants

The study was conducted with seven individuals with ASD (mean age: 13.8 +/- 2.17 years; 7 males) matched by age and sex with seven unfamiliar TDC individuals (mean age: 13.6, +/-

1.81 years; 7 males). For logistical reasons related to the availability of the device, we were not able to recruit female participants at this stage of the research. However, this limitation is compatible with the hypotheses of this pilot study; specifically, that independent of the statistical representativeness of the study, ASD individuals would exhibit differences in SOD compared to TDCs on semiecological Double Mirror paradigm. Nevertheless, we successfully matched our male participants with respect to age and age at testing.

The characteristics of the ASD and TDC individuals are presented in Table 1.

Individuals with ASD were recruited from Brest University Psychiatric Hospital in a clinic specializing in the diagnosis and assessment of ASD. The inclusion criteria were (a) a diagnosis of ASD made by a psychiatrist according to the DSM V [8] and ICD-10 [45] criteria and confirmed, at least, by Autism Diagnostic Interview—Revised (ADI-R) [46] ratings; (b) aged above 10 years old to ensure that a stable body schema had been acquired [47]; (c) absence of intellectual deficit confirmed with psychometric assessment; and (d) presence of language to permit the oral responses required in the task. We excluded individuals with (a) a history of epilepsy, (b) claustrophobia, (c) achluophobia, or (d) the presence of distinctive signs or visual deficits requiring vision correction (eyeglasses or contact lenses).

As part of routine practice in the diagnostic unit, each participant with ASD was also administered a series of clinical assessments: the ADI-R [46], the Autism Diagnostic Observation Schedule (ADOS) [48] and a psychometric assessment. One subject could not be assessed with the ADOS because he was too anxious during the assessment.

The TDC individuals matched with the ASD patients were recruited in the Brest area by oral communication with local schools via participants and staff working at the hospital. They were unfamiliar with individuals in the ASD group to avoid confusion between self-identification and familiarity. They were determined to (1) be free of any significant developmental, neurological or psychiatric disorders based on a medical examination and (2) attend typical schooling for which their chronological age corresponded to their developmental age. Similar to the ASD group, the exclusion criteria for the TDCs were (1) a history of epilepsy, (2) claustrophobia, (3) achluophobia, and (4) the presence of distinctive signs or visual deficits requiring vision correction (eyeglasses or contact lenses).

The study protocol was approved by an ethical standards committee and performed in accordance with the ethical standards laid down in the Declaration of Helsinki. Written informed consent was obtained from all participants and their parents prior to their inclusion in the study.

## Paradigm

In this experiment, we used a new double mirror paradigm based on the Alter Ego System, which consists of a semitransparent double mirror (70 cm × 50 cm × 0.4 cm; height×width×depth) with a set of computer-controlled white light-emitting diodes (LEDs) fixed on the mirror frame on both sides (Fig 1A).

These sets of LEDs can emit continuous lighting at different intensities, either separately (i.e., LEDs turned on for only one side of the mirror) or simultaneously (i.e., LEDs turned on for both sides of the mirror). Both the sampling switch between the two LED sets and the

**Table 1. Characteristics of participants.**

|  | N | Mean age | SD | Range |
|---|---|---|---|---|
| ASD | 7 | 13.8 | 2.17 | 11.5–16.8 |
| TDC | 7 | 13.6 | 1.81 | 11.5–16 |

flicker frequency range (1–20 Hz) were controlled by a PC using E-Prime software. This system enables the generation of different self-face and other-face perceptual conditions when two individuals, A and B, are facing either side of the mirror (Fig 1B):

- If the LEDs are turned on for subject A's side, whereas the LEDs are turned off for subject B's side, subject A can see his or her own face reflected in the mirror without seeing subject B's face through the mirror. This perceptual condition is referred to by Thirioux [43] and Keromnes [44] as the *self condition*.

- Using this same lighting mode, subject B can see subject A's face through the mirror (through a transparent window) without seeing his or her own reflection. This perceptual condition is referred to as the *other condition* [43, 44].

- When both sets of LEDs are on, the reflections of subject A's and subject B's faces merge in the mirror, making it potentially difficult for an individual to recognize his or her own face. The higher the light intensity is, the more visible the image of an individual in the mirror.

The experimental setting is described in Fig 1.

## Protocol and task

**Procedure.**   The experimental procedure had a duration of approximately 45 minutes.

This experimental protocol was inspired by the one previously used by Foucaud Du Bois-gueheneuc in Alzheimer's disease patients and by Gaelle Keromnes and Sylvie Tordjman in patients with schizophrenia [44]: the light intensity of the two LED sets were gradually and independently increased or decreased on the two sides of the mirror, such that the two individuals found themselves alternatively in the *other condition* or the *self condition*, as explained below.

At the beginning of the experiment, the patient was in the *other condition*, i.e., he starts by seeing the control subject's face through the mirror without seeing his own face; then, the patient's own image in the mirror becomes increasingly more apparent as a function of the light intensity.

Then, the patient is in the *self condition*, i.e., he begins to see his own face reflected in the mirror without seeing the control's face; then, with changes in the light intensity, the control subject's image in the mirror becomes increasingly more apparent.

At the same time, the control subject undergoes the same procedure, except that he starts in the *self condition* which transforms into the *other condition*.

The simultaneous alterations in light intensity on each side of the mirror experienced by the ASD patient and his matched control are presented in Fig 2A. Importantly, since modifications in the light intensity on one side of the mirror modify its reflective properties on both sides, we should consider the relative light intensity that is actually perceived by participants on each side, as shown in Fig 2B.

Similar to Keromnes et al. [44], we called a *"passage back and forth"* the series of conditions that permitted each subject to return to the starting conditions after the conditions with identical light intensities was presented twice. Thus, one passage back and forth consists of the following sequence:

| One passage back and forth | TDC progressively switch from *self to other* and then *other to self* |
|---|---|
| | ASD simultaneously switch from *other to self* and then *self to other* |

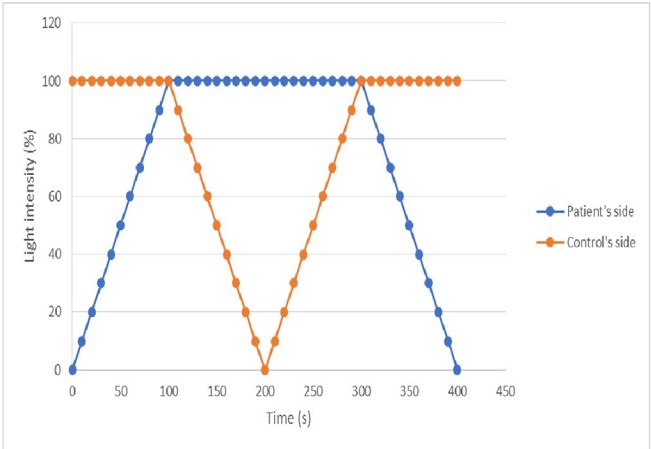

## a. Light intensity over time

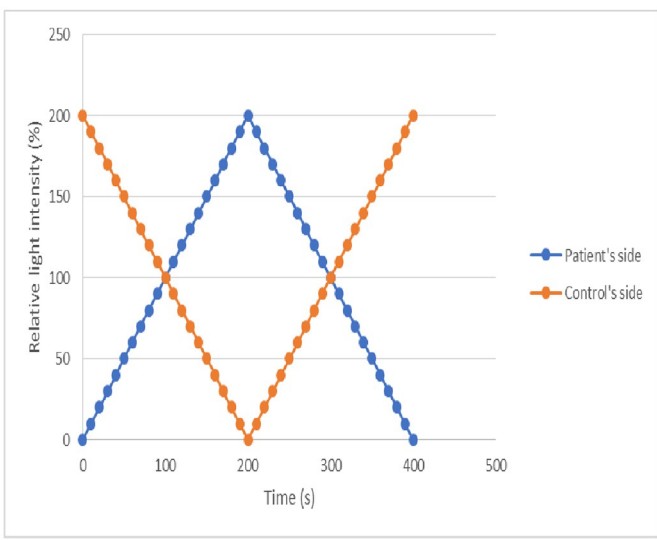

## b. Relative light intensity perceived by each participant over time

**Fig 2. Simultaneous alterations in light intensity on each side of the mirror over time.** a. Light intensity over time. At the beginning of the procedure, the light intensity was 100% for the TDCs (mirror effect), whereas the light intensity was 0% for the ASD patients (transparent window effect). Then, the light intensity is progressively increased in 10% steps on the ASD patient's side while remaining at 100% for the TDCs until 100% is reached on both sides. During this first step, the ASD patient's image progressively appears and becomes increasingly fused with the TDC's image. When both sides are at 100%, the light intensity is progressively decreased in 10% steps on the TDC's side, while it remains at 100% on the ASD patient's side. During this step, the TDC's image progressively fades until its total disappearance when the light intensity drops to 0%. Then, the reverse procedure is used to return to the initial configuration (100% on the TDC's side, i.e., a mirror effect for the TDCs, and 0% on the ASD patient's side, i.e., transparent window effect for the ASD patients). Therefore, a condition with identical light intensities is presented twice to the participants during this sequence called one "passage back and forth". b. Relative light intensity perceived by each participant over time. Relative light intensity perceived by the participants taking into account that modifications of light intensity on one side modifies its reflective properties on both sides.

The whole task consisted of two passages back and forth, i.e., the light conditions shown in Fig 2 were presented twice to each participant.

After every change in light intensity, the participants were instructed to identify whether the presented image mostly corresponded to their own face or to the other's face. Both subjects were asked a simple question: "who do you recognize most in the mirror?" The expected response was either "me" or "him/her" (to designate the other person), without any other alternative (i.e., there were no different answers and no answer was not permitted). The question was always addressed first to the individual with ASD and then to the TDC participant. Each stimulus was presented for 10 s each during a progressive sequence. The response time was unlimited. Once the answer was given by each participant, the next stimulus was presented. The type of response measured and recorded was a verbal response. The use of response button boxes was initially discussed as a way to control for bias in participants' verbal responses. However, in addition to the difficulty of using manual buttons in totally dark lighting conditions, visual control might have been required to press the button, and thus, attention to the mirror image could have been disrupted or the movement could have induced a misalignment of the 2 faces. In addition, as in Keromnes [44], oral responses seemed to reassure these patients, as the task involved maintaining relationships with others and were more relevant in terms of identity reinforcement and assertive self-recognition (using language and designation pronouns), which are central issues of this work.

The whole task consisted of two passages back and forth. There was a 10-min pause between the first and second passages to allow attentional recovery.

Debriefing occurred at the end of the experiment during an informal and friendly period of time that did not include data collection. A snack with cakes and drinks was offered during this period. The participants were asked about their feelings regarding the task and their awareness about differences in stimulation, asked whether they found face recognition and discrimination difficult, and whether they were familiar with this type of morphing (as some teenagers regularly use morphing applications).

**Data analysis.**   The main outcome was the light intensity levels at which critical changes in self/other identification occurred:

1. Level M1 was the critical perceptual threshold corresponding to the first time the individual recognized himself when his own image progressively appeared in the mirror during the *other-to-self* morphing sequence.

2. Conversely, level M2 was the critical perceptual threshold corresponding to the first time the individual recognized the other's image in the mirror when this image progressively appeared in the mirror during the *self-to-other* morphing sequence.

To summarize, the lower the M1 or M2 levels were, the larger the proportion of self was in the image (and the smaller the proportion of the other), which reflected a difficulty for the individual to uncouple from his own image.

Determination of M1 and M2 levels as well as a synthesis of the experimental procedure are shown in Fig 3.

**Statistical analysis.**   After each step (change in light intensity), the participant's verbal responses were recorded. The analysis of the main outcome variable was conducted by comparing the M1/M2 levels (expressed as percentage of light intensity) between individuals with ASD and TDCs. As this variable was not normally distributed, the nonparametric one-tailed Mann–Whitney test was used to compare M1 and M2 levels between the two groups. To assess potential relationships between performance on our task and the severity of ASD, we computed correlations of M1 and M2 levels with ADOS severity scores. Since the data were not

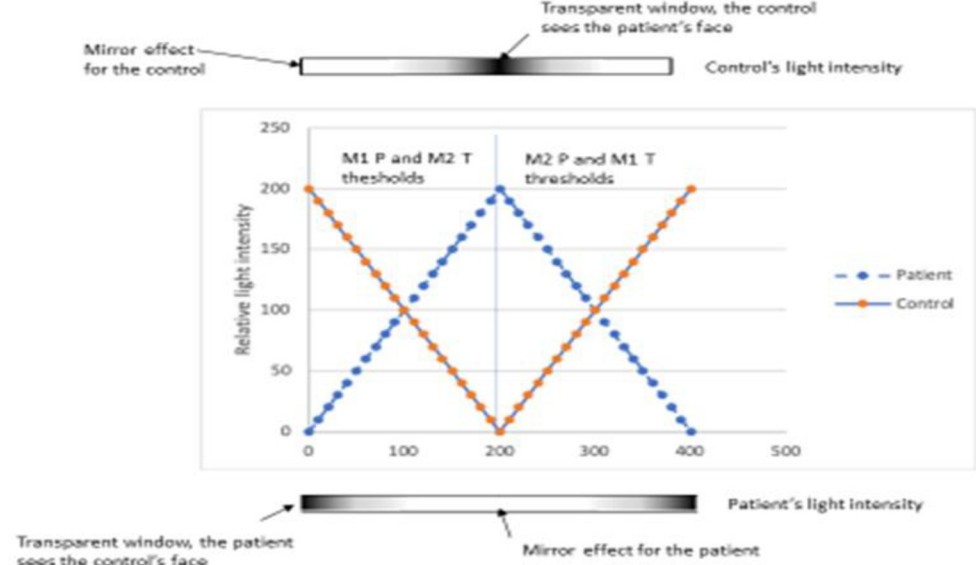

**Fig 3. Synthesis of the experimental procedure during one "passage back and forth".** During the task, the image the ASD patient sees undergoes a morphing from other to self and then from self to the other. Simultaneously, the image the TDC sees undergoes a morphing from self to other and then from other to self. The threshold M1 corresponds to the light intensity at which the perceptual shift from other to self occurs. The M2 threshold corresponds to the light intensity at which the perceptual shift from self to other occurs. The procedure is then repeated a second time. Thus, for each subject, 2 M1 and 2 M2 thresholds are obtained.

normally distributed, we used Spearman's correlation analyses; the associated p-value, which reflects whether the dependency is statistically significant, was computed using the algorithm AS 89 (see [49]) implemented in R.

## Results

The results for the mirror SOD task in individuals with ASD and TDCs are presented in Table 2.

M1 levels were significantly lower in the individuals with ASD than in the TDCs ($p = 0.001$). This indicated that the ASD individuals showed an "earlier" self-recognition in the *other condition* (i.e., when one's own image was gradually appearing in the mirror in the *other-to-self* direction of the morphing).

Similarly, M2 levels were significantly lower in the ASD individuals than in the TDCs ($p = 0.023$). This indicated that the individuals with ASD showed a delay in recognizing the

**Table 2. Comparison of the results for the recognition task between individuals with ASD (N = 7) and typically developing controls (N = 7) for the first passage back and forth (M 1st), the second passage back and forth (M 2nd) and mean value (M).**

|  | Individuals with ASD (N = 7) | | | Typically developing controls (N = 7) | | |  |
|---|---|---|---|---|---|---|---|
|  | Mean | Median | SD | Mean | Median | SD | P value |
| M1 1st | 62.86 | 60 | 37.29 | 145.7 | 140 | 15.11 | 0.002 |
| M1 2nd | 87.14 | 90 | 39.88 | 144.3 | 140 | 9.75 | 0.003 |
| M1 | 75 | 70 | 36.1 | 145 | 145 | 8.2 | 0.001 |
| M2 1st | 127.1 | 120 | 19.76 | 150 | 160 | 22.36 | 0.015 |
| M2 2nd | 112.9 | 120 | 29.94 | 137.1 | 140 | 24.98 | 0.07 |
| M2 | 120 | 125 | 20.4 | 143.6 | 150 | 22.1 | 0.023 |

other in the *self condition* (i.e., when the image of the other was gradually appearing in the mirror in the *self-to-other* direction of the morphing).

Moreover, there was greater interindividual variability in M1 thresholds among the ASD patients than among the TDC patients. This dispersion of the data is visible by the standard deviation for the ASD patients compared to the TDCs and possibly due to heterogeneity within the ASD group. Notably, subject A7 showed M1 scores of 130 in both the first and second passages back and forth, while the M1 average score for the other ASD individuals during the 1st passage back and forth was 51.6. Further details regarding this subject are provided later in the discussion (see below).

For the ASD group, Wilcoxon paired-samples tests showed a significant increase in M1 and M2 thresholds between the first and second passages back and forth (*p = 0.046 and 0.931, respectively*). This indicated that the M1 and M2 recognition thresholds moved closer to those of the TDCs on the second passage back and forth. This difference could indicate that the sensitivity of the ASD subjects on the task improved over time, but additional testing would have been needed to confirm this.

The Wilcoxon paired-samples test comparing M1 and M2 showed that M2 levels were significantly higher than M1 levels in individuals with ASD (*p = 0.0342*). Conversely, there was no significant difference between M1 and M2 levels in TDCs (*p = 0.6707*). This indicated that, in the TDCs, the critical perceptual threshold corresponding to the ability to recognize self and other was the same regardless of the direction of morphing. In other words, the critical threshold for switching between self and other appeared when viewing morphed faces that contained the same proportion of facial features of self and other, regardless of whether it was the *other condition* or *self condition*. In contrast, in the individuals with ASD, this perceptual threshold varied depending on the direction of the morphing, and the shift from self to other or from other to self did not occur at the same threshold. Indeed, the ASD group needed a smaller proportion of self (image of themselves) to recognize their own face in the *other-to-self* direction versus the *self-to-other* direction.

A synthesis of the distribution of M1 and M2 levels is presented in Fig 4.

To assess potential relationships between performance in our task and autism severity, we calculated correlations between the M1 and M2 thresholds and the ADOS severity scores. The results are summarized in Table 3.

Fig 5 summarizes the relationships among the different variables. We found a positive correlation between M1 and M2 (in blue) and a negative correlation of M1 and M2 with ADOS scores (red). The lowest p-value was found regarding the relationship between the M1 threshold during the second passage back and forth and the overall ADOS score (*correlation coefficient -0.79, p = 0.059*), indicating earlier face recognition when one's own image gradually appeared in the more severely affected ASD individuals. M2 levels and ADOS scores were not significantly correlated. Regarding strongly-correlated ADOS sub-scores, a slightly lower p-value was found in the relationship between the M1 level and the 'interaction' score compared to the 'communication' score. Therefore, the ADOS interaction score is most closely linked to M1.

## Discussion

Our results are based on a very small, male-only sample; therefore, the preliminary nature of this study must be taken into account when analysing and discussing the results.

### Comparison with previous research using static images

Our results showed that the TDCs and ASD individuals tended to recognize their own images above the 50% self/other level of illumination (i.e., the objective "switch" point between self

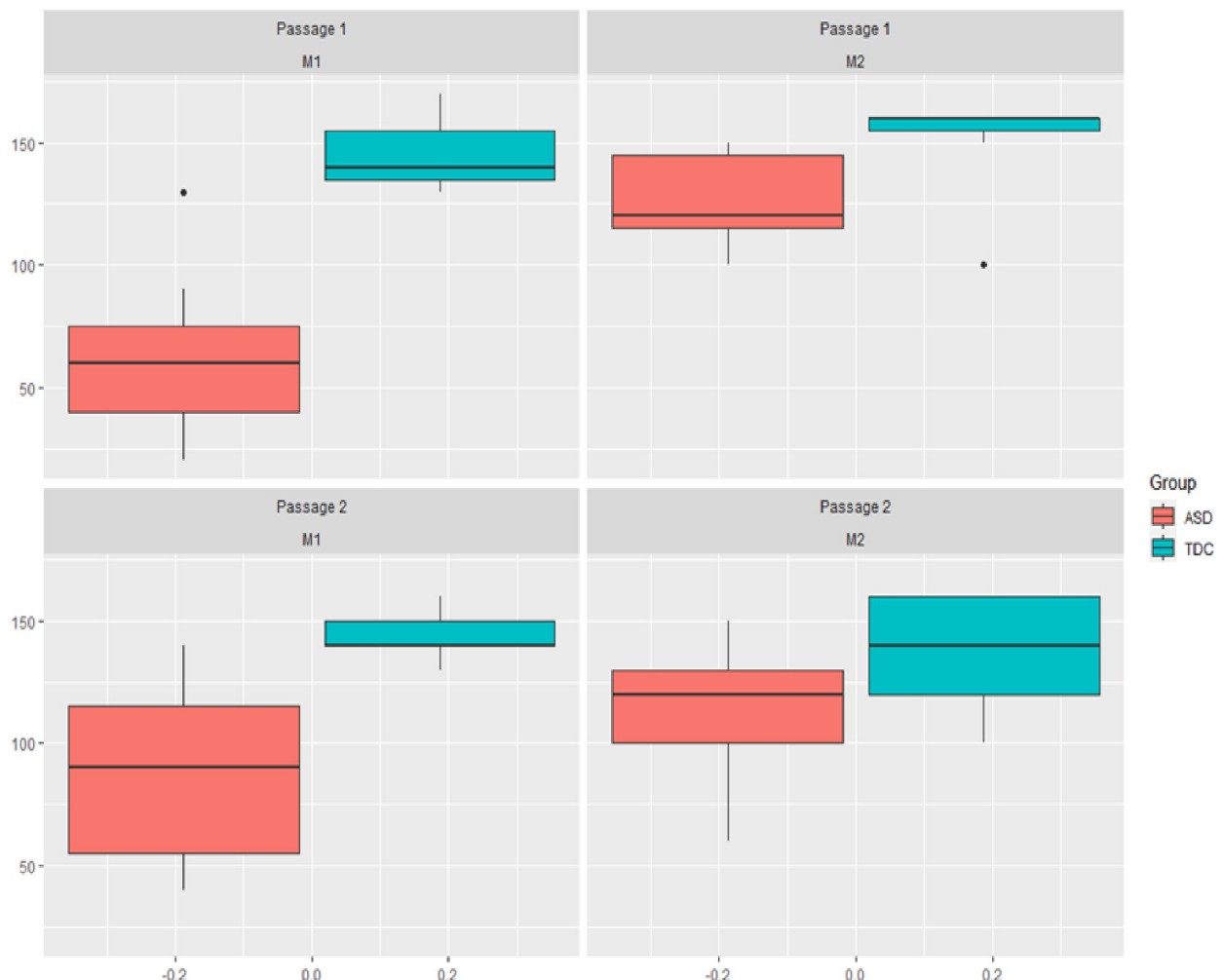

**Fig 4. Distribution plots of M1 and M2 during the first and second passages back and forth for ASD individuals and TDCs.**

and other) in the *self-to-other* direction, while in the *other-to-self* direction, the participants tended to recognize themselves slightly below the 50% self/other level. These findings contrast with SOD tasks using static images showing that healthy subjects [22–24] and ASD individuals [27] stopped recognizing themselves slightly before the 50% self/other point in the *self-to-other* direction, while in the *other-to-self* direction, participants tended to recognize themselves slightly after the 50% self/other point. This discrepancy is consistent with Thirioux's findings

**Table 3. Comparisons between M1 and M2 levels and ADOS scores in ASD individuals (N = 7).**

| | M1 | M2 | ADOS-2 scores | | |
| | | | Reciprocal social interaction item | Communication item | Total |
|---|---|---|---|---|---|
| A1 | 70 | 135 | 8 | 3 | 11 |
| A2 | 50 | 85 | 9 | 4 | 13 |
| A4 | 35 | 125 | 8 | 4 | 12 |
| A5 | 115 | 135 | 4 | 3 | 7 |
| A6 | 80 | 100 | 2 | 2 | 4 |
| A7 | 130 | 120 | 4 | 3 | 7 |

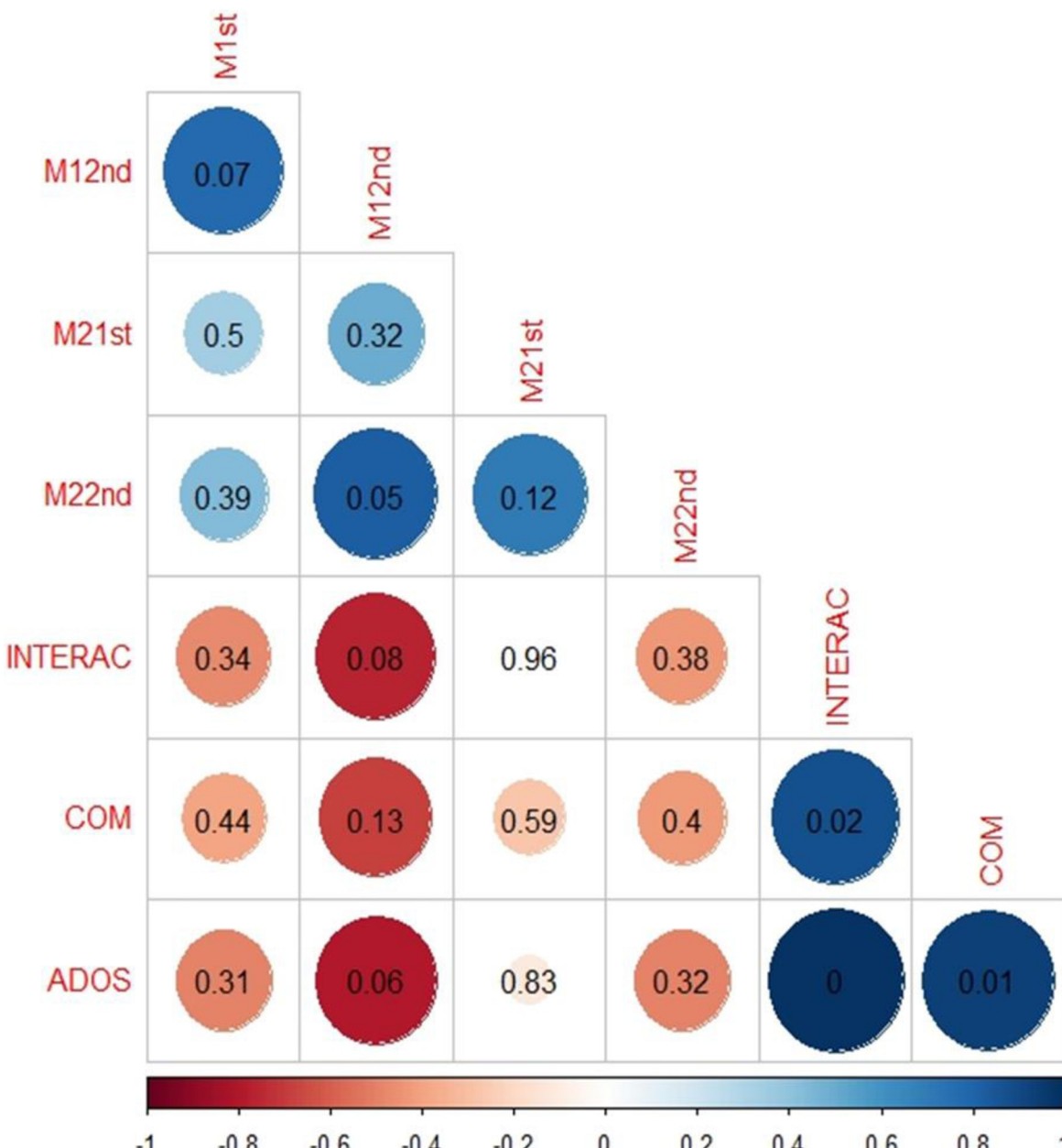

**Fig 5. Spearman's correlation analyses of different pairs of variables (various colours) and the associated p-value (indicated by the text).**

using a double mirror with healthy subjects and confirmed that static images and double mirrors are different paradigms [43].

In addition, our results confirmed that, by comparison with the TDC subjects, the ASD patients might have had problems in SOD in two directions: (i) in the *self-to-other* direction of the morphing, the shift from "self" to "other" showed an egocentric bias (i.e., continued to see "self" later than the TDCs) and (ii) in the *other-to-self* direction, the shift from "other" to "self" also showed an egocentric bias (i.e., started to see "self" earlier than the TDCs). Furthermore,

SOD deficiencies showed a trend-level correlation with ASD severity, with earlier self-face recognition performance in more severely affected individuals.

The differences between the ASD individuals and TDCs in the mirror condition, which contrasts with the lack of significant differences in previous research using static images [27], confirmed an interest in paradigms involving online interactive tasks that increase the amount of contextual self- and other-related information.

## Egocentricity in SOD

**Overinclusion of the other's face.** *Comparison between the self and other conditions.* Our results showed that the ASD individuals, compared to the TDCs, judged the morphs to be more like the self in both directions of morphing. Importantly, the comparison between self and other conditions shows that, unlike the TDCs, the ASD individuals switched from other to self and from self to other at different thresholds. This discrepancy indicated that the confusion between self and other may have been determined by a unidirectional overinclusion of the other's attributes by the ASD individuals. Conversely, the TDCs may have exhibited compensatory mechanisms with bidirectional self- and other-facial feature attribution. Indeed, they overattributed their facial features to the other's face and overattributed the other's facial features to their own face. This understanding is supported by Thirioux's findings using a double mirror in healthy subjects [43] and human interpersonal understanding based on balanced mechanisms of self-projection and simulation [5, 6, 50].

*Implications for sensory processing.* Our findings can be interpreted as a greater level of assimilation of the other's face in the representation of one's own face among the ASD individuals than the TDCs. Such misattribution of others' facial features to oneself has been reproduced in nonclinical populations through simple psychophysical manipulations in a procedure involving sensory processing called interpersonal multisensory stimulation [51]. Indeed, studies using body ownership illusions have shown that, under certain conditions, the sense of self can be manipulated to include a fake or another person's body part, for instance, the hand [52], face [53–55] or whole body [56].

Sensory processing in individuals with ASD during social interactions is not fully understood, but there is agreement that atypical multisensory processing is a critical component of this complex neurodevelopmental disorder (for a review, see [57]). Notably, individuals with ASD have been shown to have less susceptibility to body ownership illusions, especially reduced and delayed susceptibility to the rubber hand illusion (RHI) [58–60] and full body illusion [61], than controls.

The double mirror task involves a simultaneous visual perceptual conflict between the visual input ("I see the other") and the proprioceptive experience ("I expect to see myself in the mirror") that are necessary for the identification of one's own face [53]. This conflict is reinforced by the everyday experience we may have developed that, when we look in the mirror, we see ourselves and not another person.

Our results indicated that people with ASD continued to recognize themselves in the mirror despite the increase in the proportion of the other person's face in the image, regardless of the direction of the morphing. These results may indicate that, in ASD individuals, the proprioceptive system would be less vulnerable to bias originating from visual information (i.e., ASD individuals depend more heavily on proprioceptive information than on visual information when incongruent). This finding is in line with previous studies that pointed out an overreliance on proprioception in the presence of competing signals from other modalities [62].

Importantly, reduced sensitivity to embodiment illusions in ASD is generally interpreted as reflecting a weaker tendency to incorporate the nonself into the self (i.e., difficulties in

embodying the bodily other), consistent with ASD being associated with a greater gap or steeper gradient from self to other [63]. Our results, by contrast, support an enhanced embodiment illusion in the double mirror task (i.e., a reduction in SOD).

This discrepancy may be because our experimental design differed from that of previous studies because we investigated the effects of a unisensorial *(visual)* conflict, while prior studies investigated the effects caused by multisensory *(visuotactile)* illusions [53, 54]. Interestingly, our results are in line with a very recent study showing a greater susceptibility to embodiment in ASD individuals during a nonvisual (i.e., unimodal) variant of the RHI [64].

Moreover, in contrast to a greater gap between self and other [61], our results supported the notion of a reduction in SOD, i.e., difficulties in distinguishing the self from others on a sensorimotor level. This hypothesis is in line with an EEG study using an action-based somatosensory congruency paradigm showing difficulties in higher-order SOD processing based on a unimodal touch in ASD participants [65].

**Difficulties in uncoupling from one's own image.**   *Implications regarding visuospatial abilities*. The results obtained in the present study showing earlier self-recognition and delayed other recognition in the ASD individuals, compared to the TDCs, could also be interpreted as a difficulty to inhibit their own perspective during the face recognition task and adopt the reference frame of the other, resulting in a privileged use of the egocentric reference frame.

Indeed, unlike when one is faced with a simple photograph, the mirror requires processes of perspective shifting and spatial transformations beyond pure self-recognition. In particular, self-recognition in the mirror requires matching one's sensorimotor experience (1st person perspective) with the object seen in the mirror (3rd person perspective), thereby identifying the "I" with the "me" and representing that self as an object to others and to oneself [41]. Moreover, the face-to-face postural configuration of both subjects during the Alter Ego double mirror task requires a 180˚ mental rotation of one's own body. In particular, Thirioux et al. [43] emphasized the role of visuospatial abilities on self-recognition and SOD within this task.

The ability to uncouple from an egocentric point of view does require a mental transformation as well as an inhibition of one's own dominant perspective (first person perspective), possibly involving executive functioning (mainly inhibitory control), with both abilities showing developmental progression into childhood [66–68]. From a developmental viewpoint, research has shown that this ability occurs between 10 and 11 years [69, 70]. Our results in adolescents aged 11–16 years are consistent with impairments in the onset of spatial reference frame processing in the ASD group compared to the TDC group. This result is supported by previous research highlighting the notion that egocentric transformation is deficient in people with ASD [71] and is supposed to be linked with ASD symptomatology, especially high levels of autistic traits [72, 73]. Moreover, impaired inhibitory control in the ASD individuals during the double mirror task is possible; however, relevant studies have yielded contrary results in the literature (for a review see [74]) notably in the social context [75]. In addition, repeating the double mirror task with other younger and older groups could help clarify the developmental age effects that may be involved.

*Comparison between the first and second passages back and forth*. Our results showed a later self-recognition in the second passage back and forth for the individuals with ASD compared to the first passage. This indicated that, during the second passage, the threshold for self-recognition approached that of the TDCs. This finding suggests that changing perspective is possible, although delayed, in ASD. This time-dependent difference in adaptability might be typical for changing perspectives in a spatial environment and conforms to previous behavioural studies employing comparable mental own-body transformation (OBT) tasks with ASD participants (i.e., egocentric versus allocentric) [76]. The assessment of additional passages back and

forth would have been necessary to validate this hypothesis but could have been excessively tiring for participants.

**Impairments in switching between the abstracts of self and other.**   Our experimental design requires switching from *self to other* and from *other to self* as the morphing gradually shifts from 0% to 100% between self and other and vice versa. Thus, the double mirror Alter Ego task is consistent with recent conceptualizations of SOD as being achieved through SOC, i.e., the ability to switch between representations of self and other and to inhibit the representation that is not relevant in a given situation.

In our study, differences in M1 and M2 thresholds between ASD individuals and TDCs were consistent with ASD being associated with an inadequate control of the self/other switch [77]. In particular, our results indicated that the ASD individuals continued to recognize themselves later in the *self-to-other* direction and begin to see themselves earlier in the *other-to-self* direction than the TDCs. These results are consistent with the fact that the default state of the self/other switching process is self and that moving from self to other is an active process that requires some effort [21, 78]. This ability to shift from self to other, also involved in directing attention to others, is thought to be significantly impaired in ASD, consistent with their lower social attention [79].

## Self-recognition in the mirror

**Implication for bodily self-consciousness.**   Self-consciousness allows both self-recognition and differentiation between self and other and is the basis for social interactions [49]. In line with previous research showing the value of the double mirror device in studying bodily self-consciousness [44, 80], our results support the notion that self-disorder is an important dimension of ASD [81, 82] and, in particular, that the embodied aspects of the self could be disturbed [61, 83].

This hypothesis is consistent with the fact that the sense of self related to body image development seems to be lacking in children diagnosed with ASD from an early stage [84]. Notably, face recognition in the mirror is demonstrated by typically developing children at approximately the age of two, and many young children with ASD show a delay in the development of this ability [85], as well as a delay in the use of personal pronouns ("I" and "me") [86].

Interestingly, using the same protocol in patients with schizophrenia versus healthy controls, Keromnes and Tordjman [44] found similar results to ours (i.e., earlier self-recognition and later other-recognition). From a developmental perspective, this supports the idea of a continuum between autism and schizophrenia spectrum disorders [87, 88] and, in particular, that disturbances in body image development that are present from infancy may be a shared dimension of these disorders in which there are problems of nondifferentiation of the self and consequent problems in the development of social communication [89].

**Combining egocentricity in SOD and bodily-consciousness impairments.**   To summarize, our findings highlight (i) disturbances in mirror self-recognition suggesting bodily self-consciousness impairments and (ii) egocentric bias in SOD found among the ASD participants compared to the TDCs.

These results are consistent with authors interested in SOD impairments who underscored a paradox in ASD that combines egocentricity with a weakened sense of self [11, 90].

However, this paradox is apparent only if we consider that different key dimensions of bodily self-consciousness that are closely interconnected could be impaired in ASD, leading to disruption of several phenomenological and cognitive aspects of the self. Notably, on the one hand, multimodal sensory processing and integration are used to build the basis of the self (self-identification) [91, 92], and on the other hand, the integrity of the bodily self is necessary

to change the reference frame through self-location and transformation [73, 93]. Thus, impairments in bodily self-consciousness are consistent with the higher egocentricity we found in the double mirror task involving both multisensorial and visuospatial dimensions.

In keeping with the phenomenological tradition of the "specular double," the participants with ASD behaved as if they were more likely to see a "second self" on the other side of the double mirror. Such subjective experiences of illusory duplication of the self are also found in clinical observations [94] or autobiographical accounts of individuals with ASD [95, 96]. These so-called "out-of-body experiences" (OBEs) are related to disturbances of the temporoparietal junction [97], an area specialized in various self-referential processes, notably in the SOD [19, 98]. In particular, self–other facial morphing tasks have shown that inhibition of this area resulted in participants judging the morphs to look more like the self (i.e., egocentric bias) [99], as we found in the ASD participants. Notably, various functional or structural alterations of this brain area have been reported in people with ASD [100].

## Implications for understanding ASD

**From perceptual SOD to other levels.**   The self–other facial morphing task used in this study taps into *perceptual* SOD, that is, the capacity to identify one's own body (here, one's face) and to distinguish it from others. Therefore, it raises the question of whether the findings from the present study could be generalized to other domains of SOD. Although perceptual SOD and mental-state SOD should not be equated, evidence shows that they may be related [4, 91] and that SOD may operate in a domain-general rather than domain-specific manner [19]. Thus, the results of the present study support the hypothesis that individuals with ASD may have early impairments in the low-level components of SOD that involve a clear physical distinction between self and other. This impairment may then disrupt the development of more complex ASD skills, such as attribution of intentions and other mental states and imitation [101].

**Therapeutic and research implications.**   Our results highlight possible therapeutic implications of the double mirror Alter Ego methodology.

Indeed, a significant increase in the self-recognition threshold (M1) was found during the second passage compared to the first passage, which could reflect a learning phenomenon. Additionally, one ASD participant (A7) explained during the debriefing time that he found the task very easy because he was very familiar with video game morphing sessions. His results showed M1 scores comparable to those of the TDCs.

These results are particularly interesting because previous research has shown that training participants to control motor representations of self and others can improve their ability to control imitative behaviours and better adopt the visual perspective of others [102]. This suggests that SOD training at the perceptual level could then transfer to another social domain.

In addition, a unique role of visual perspective in the mirror view for SOD was demonstrated in somatoparaphrenic patients. When patients saw their hand directly, they attributed ownership of the limb to someone else, whereas seeing the limb through the mirror reflection allowed them to claim it as their own [103]. The therapeutic value of the mirror is also consistent with observations that regular exposure of children with developmental disabilities to mirrors is associated with overall positive behavioural change [104].

These findings suggest a potential remediation pathway using double mirrors by practising control of representations of self and others and simultaneously improving the sense of body ownership, particularly during the period of self-construction. However, remediation could go beyond that as well. Indeed, recent research has shown that adult women with ASD who exhibited neurotypical patterns of activation during self-presentation (as opposed to other-

presentation) were also better in camouflaging (i.e., behaviourally acted like neurotypical individuals) [105], suggesting that SOD abilities may be important for compensatory skills in ASD. Interestingly, subject A7, who had M1 scores close to those of the controls, had low-severity ASD with one of the lowest ADOS scores among the participants.

## Limitations

Some limitations of the study should be recognized. First, this is a preliminary study conducted with very small, male-only samples. Future studies are therefore needed to replicate the results in larger, mixed samples.

In addition, there might be a bias of participants' oral responses influencing each other. To decrease this possible bias in the individuals with ASD, these participants were consistently asked to respond first during the task. Aleatory variations in light intensity rather than progressive linear changes were initially discussed to control for possible habituation bias in participants. However, abrupt changes in light intensity and the reflected images could have been stressful for the participants. Moreover, in line with other studies [24, 53], we presented the morphs incrementally from 0% self/other to 100% self/other in the two directions separately. This allowed us to differentiate the morphing directions and disentangle the two types of self/other confusion (i.e., egocentric bias and altercentric bias) and the critical thresholds for switching between self and other as required by the SOC process.

Similarly, the order of the presentation of the task was not randomized, which could have led to carry-over effects. However, there was no significant time effect on the results of the recognition task in the individuals with ASD or the TDCs, which allowed us to reduce the possible carryover and learning effects of the task.

Another limitation of this study is that the differences between ASD individuals and TDCs may have occurred at the attentional or information processing level, i.e., upstream of the SOD process. Significantly, research has shown differences between ASD individuals and healthy controls in the pattern of eye fixation during face recognition [106], and visual information from the eye region contributes to recognition abilities [107], including decisions about face identity [108]. Moreover, comorbidities in the ASD participants, such as social anxiety or ADHD symptoms, could have played moderating roles being unable to extract eye information or leading to information processing issues, respectively. These differences in eye fixation patterns may require the use of an eye-tracking device in the double mirror task to control for this bias. Another hypothesis concerns the attentional processes involved in our task, in particular the self-reference effect (SRE), which assumes that encoded information about the self provides a mnemonic advantage over information encoded in other ways [109]. The predominant egocentricity during the double mirror task could then be related to a higher SRE in the ASD individuals than in the TDCs. However, several studies have shown either a reduced or typical SRE in those with ASD [110]. Second, the instructions used in our protocol did not require stimulation of the SRE. Indeed, unlike other studies about self/other recognition that focused on detection of the self (and likely decreased the amount of attention directed towards others), our instruction was "who do you recognize most in the mirror?". However, future work should examine whether a simple change in the task instructions (e.g., asking participants to press a button when they detect the presence of themselves) may change the results obtained in the present study.

Finally, much research contrasts local versus global processing, with a bias towards local processing in individuals with ASD [111]. This could explain the differences based on morphing direction (*self to other* or *other to self*), which was found only in the participants with ASD. Indeed, in participants with ASD, both local and global processing could have been engaged to

achieve the same performance, regardless of the direction of morphing. Conversely, preferential local processing might have allowed the ASD subjects to recognize their own face earlier than the TDCs during the morphing sequence from *other to self* (focusing on details belonging to the self, according to the SRE effect). However, this local processing could not have compensated for the lack of global processing when unfamiliar facial features appeared in the other direction.

## Conclusion

Although further research is needed to replicate these results, this study uncovered novel findings showing the first behavioural evidence of impaired SOD in individuals with ASD through an embodied face-recognition paradigm generating a self–other face merging illusory effect in ecologically relevant conditions, i.e., when two individuals are physically facing each other and interacting.

Our results support the hypothesis that the inability to correctly distinguish the self from the other and to switch from one representation to another may play a key role in clinical and subclinical symptomatology observed in those with ASD. Furthermore, the use of a mirror (i.e., a device known to be a key marker of the self in early development) suggests that those SOD impairments could be related to early deficits in the development of bodily self-consciousness.

Finally, similar results found in previous research using the same paradigm in schizophrenia patients raise the question of considering SOD disturbances as a transdiagnostic dimension shared with other psychiatric or neurodevelopmental disorders and support the interest of complementary research using the double mirror Alter Ego at the diagnostic but also therapeutic level.

## Acknowledgments

The authors thank Pierre Ailliot and Cyril Dusausoy for the statistical analysis used in the study.

## Author Contributions

**Conceptualization:** Nathalie Lavenne-Collot, Marie Tersiguel, Michel Botbol, Alain Berthoz.

**Data curation:** Nathalie Lavenne-Collot, Marie Tersiguel, Céline Degrez.

**Formal analysis:** Nathalie Lavenne-Collot, Marie Tersiguel.

**Investigation:** Marie Tersiguel, Céline Degrez.

**Methodology:** Nathalie Lavenne-Collot, Nolwenn Dissaux, Michel Botbol, Alain Berthoz.

**Project administration:** Nathalie Lavenne-Collot, Michel Botbol, Alain Berthoz.

**Supervision:** Nathalie Lavenne-Collot, Alain Berthoz.

**Validation:** Michel Botbol, Alain Berthoz.

**Writing – original draft:** Nathalie Lavenne-Collot, Nolwenn Dissaux, Michel Botbol.

**Writing – review & editing:** Nathalie Lavenne-Collot, Nolwenn Dissaux, Guillaume Bronsard, Michel Botbol, Alain Berthoz.

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
