## [Decision Letter · Decision Letter 0]

13 May 2022

PONE-D-22-04219Self/other distinction in Autism Spectrum Disorder (ASD) studied in adolescents with a double mirror paradigmPLOS ONE

Dear Dr. Nathalie Lavenne,

Thank you for submitting your manuscript to PLOS ONE. After careful consideration, we feel that it has merit but does not fully meet PLOS ONE’s publication criteria as it currently stands. Therefore, we invite you to submit a revised version of the manuscript that addresses the points raised during the review process.

We look forward to receiving your revised manuscript.

Kind regards,

Mariella Pazzaglia

Academic Editor

PLOS ONE

Journal Requirements:

Additional Editor Comments (if provided):

We have received our consultants' reports on your manuscript. As usual, I have invited two experts in your field of research, whose comments you will find below.

As you will see, neither of the reviewers is really enthusiastic. Part of the problem is many inaccuracies and shortcomings that the reviewers discuss in their excellent, detailed reports. These criticisms coincide with my own impressions. On the one hand, this means that there is still much work to be done to improve this manuscript. On the other hand, however, I have not seen any dramatic flaws in the design, which means that these efforts may indeed be worthwhile. I would therefore ask you to prepare a comprehensive revision that attempts to address all remaining concerns as much as possible, along with a cover letter that addresses all concerns step by step. My plan is to send the revision to the same two reviewers if possible. Please note that I have not yet committed to a final judgement

Reviewers' comments:

Reviewer's Responses to Questions

**Comments to the Author**

1. Is the manuscript technically sound, and do the data support the conclusions?

Reviewer #1: No

Reviewer #2: Yes

2. Has the statistical analysis been performed appropriately and rigorously? 

Reviewer #1: No

Reviewer #2: Yes

3. Have the authors made all data underlying the findings in their manuscript fully available?

Reviewer #1: No

Reviewer #2: Yes

4. Is the manuscript presented in an intelligible fashion and written in standard English?

Reviewer #1: Yes

Reviewer #2: Yes

5. Review Comments to the Author

Reviewer #1: This was an innovative study design using a unique "Alter Ego" double mirror paradigm that provided an examination of self-recognition and self-other distinction processes among adolescents with ASD. While this is an important and interesting topic, there are several major concerns that need to be addressed.

First, the Introduction and Discussion, while very comprehensive, should be shortened and more concise. For example, the section on neurobiological findings in the Introduction (page 8) could be removed. In the Discussion, only literature specifically related to this study’s findings should be included.

Second, the paper overall needs to be carefully proofread for grammatical issues. There are numerous spelling errors throughout. Also, decimal points should replace the commas in all numbers.

Third, there are few concerns/questions regarding the Materials and Methods section:

-The participants should be described more clearly, including the inclusion/exclusion criteria. Why were all participants male and between the ages of 11 and 17? Were females attempted to be recruited? Why not younger individuals as discussed later in the Discussion section (page 25)? Why did one individual not receive an ADOS assessment? This isn’t then mentioned when the correlation coefficients are assessed. The TDCs were recruited by participants and staff, but where/how were they recruited?

-On page 15, the first point of the Figure 2a description states that both the ASD patients and TDCs start at 100% light intensity. I believe this should be 100% for TDC and 0% for ASD?

-The task consisted of two passages back and forth, both starting with the ASD individual at the other condition and the TDC at the self condition. Why wasn’t one pass started with the TDC at the self condition and then the TDC starting at the other condition at the second pass. In other words, I feel that possibly starting the ASD individual at the other condition each time may possibly create some sort of “bias.” I understand that they both go through each “phase” the same amount of times, but just how the process begins seems like it may impact the outcomes.

-There are also come ambiguous/subjective comments in the methods (page 16). For example, “the expected response was either “me” or “he/she.” How were they told to respond, what were the actual responses, and was it possible to not get any response? Another example, “The type of response measured and recorded is a verbal response, which seems more relevant than a button response…” Why was this more relevant? Move the description of this reasoning from the Discussion to the Methods and change the wording to be less subjective. Finally, “…a systematic 10 minutes pause…”; what was systematic about it? Why 10 minutes?

-The end of the Procedure section mentions a debrief during which participants were asked different questions. Why were these questions asked and where are the results/responses?

-In general, the Data and Statistical Analysis sections needs to be expanded to include more detail, and the specific analyses should be better explained. The outcome is stated to be the light intensity levels (expressed as percentage), but it’s unclear as to how these were calculated. The results in Table 2 show percentages larger than 100. Shouldn’t these values be between 0 and 100, and only in increments of 10, since the light increased 10% each step?

Finally, the Results section includes statistical tests (page 19) that were not mentioned in the data analysis section, and the p-values presented on page 19 are not included in any of the tables and are unclear as to where they came from. Also, Table 3 of the “correlation analysis” doesn’t include any of the actual correlation coefficients.

Reviewer #2: I apologize for my belated review. I appreciate this opportunity to review Lavenne-Collot and colleagues’ novel and rigorous work of using a novel double mirror task to provide an ecologically valid paradigm to test self-other distinction problems in autism. To be honest, I personally learned a great deal from reading their work. Nonetheless, I have some comments, which hopefully may help improve the current form of the manuscript.

1. I highly respect the French language and culture. Nonetheless, there are several style issues, which are not conventionally used in the English language, throughout the manuscript. Given that this article is going to published in English journal, I suggest that these issues should better be amended. Specifically,

‘i-e’ should be ‘i.e.’

‘e.g,’ should be ‘e.g.’

The Guillemets ‘« »’ should be expressed in ‘ ” ” ’ or ' ' ' ', the quotation marks which are commonly used in English

’13,8’ should be ’13.8’, and this should also applied to all number style

2. ‘Embodiment’ was first mentioned in Introduction (Page 9), but was not explained. A concise explanation should be added.

3. In Methods and Results, autism severity used in the correlation analysis with M1 or M2 was not clearly stated. I wonder whether it is social-communication in ADOS-2 or social and communication, respectively, in ADOS-2 was used in the correlation analysis.

4. On Page 21, “In comparison with the social interactions score and then the communication score, no correlation was significant” This statement was inaccurate to me.

5. I think the biggest caveat of this study is very small sample size (n=7 in each group). Together with the limitation of male-only sample, these two caveats (male, very small sample size) should be stated very clearly in the first paragraph of Discussion and in the conclusion, in order to explicitly remind the readers that before starting discussing the results and before making the conclusive remarks, they need to be aware of these limitations, resulting in preliminary nature of this study.

6. On Page 21, “Moreover, SOD impairements were correlated with ASD severity” This statement is inaccurate, as there was only a trend-level correlation, despite a large effect size. This statement should be toned down.

7. On Page 25, the authors noted that their speculation, that response inhibition impairments in ASD may in part explain the SOD problems, is inconsistent with the Review (83). Nonetheless, this cited review was mainly based on the inhibitory control task in non-social contexts. The authors may also cite https://www.sciencedirect.com/science/article/pii/S221053361830022 to argue that the evidence of inhibitory control impairments in social contexts associated with autism is actually inconclusive.

8. I did not find which result could support the discussion on Page 27 and 28 surrounding the “disturbances in mirror self-recognition suggesting a weakened sense of self”. Could authors please clarify this part?

9. On Page 30, “These similar findings support the idea of a continuum between autism and schizophrenia spectrum disorders (37, 114, 115).” https://jamanetwork.com/journals/jamapsychiatry/fullarticle/2773832 This reference is very relevant to this statement and could be cited here.

10. On Page 30 and 31, it’s very interesting to note that one autistic participant indicated the practice and learning effect. The authors used this to support their further argument surrounding the clinical implication. I like the approach. Nonetheless, I wonder what his autistic severity was. If his autistic severity was relatively low, this can be further used to support that by training SOD using the double mirrors or other similar paradigms, people may be able to have their behaviours modified.

11. On Page 32 and Page 33, the authors discussed whether some other alternative theories/factors could explain the present results. I wonder whether social anxiety and ADHD symptoms could play a role in moderating being unable to extract eye’s information or information processing issues, respectively.

12. On Page 33, “Conversely, their preferential local processing might have allowed ASD individuals to recognize their face earlier than TDCs during the other to self morphing sequence.” This statement is unclear to me. Could authors please provide a more detailed explanation for this sentence and the argument?

6. PLOS authors have the option to publish the peer review history of their article (what does this mean?). If published, this will include your full peer review and any attached files.

Reviewer #1: No

Reviewer #2: No

---

## [Author Response · Author response to Decision Letter 0]

18 Jul 2022

Reviewer 1: 

This was an innovative study design using a unique "Alter Ego" double mirror paradigm that provided an examination of self-recognition and self-other distinction processes among adolescents with ASD. While this is an important and interesting topic, there are several major concerns that need to be addressed.

First, the authors would like to sincerely thank the reviewer for his careful proofreading of the article. We have taken into account all the remarks and correction requests to improve this work and hope that the reviewer will judge that these evolutions are sufficient to reach the standards of publication.

1). First, the Introduction and Discussion, while very comprehensive, should be shortened and more concise. For example, the section on neurobiological findings in the Introduction (page 8) could be removed. In the Discussion, only literature specifically related to this study’s findings should be included.

We thank the reviewer for this comment. The introduction and discussion have been shortened and made more concise. In the introduction, the section on neurobiological findings has been removed. In the discussion, only literature specifically related to the results of this study has been included. Moreover, the organization of the manuscript has been revised to make it more coherent and intelligible.

2.) Second, the paper overall needs to be carefully proofread for grammatical issues. There are numerous spelling errors throughout. Also, decimal points should replace the commas in all numbers.

The authors thank the reviewer for this advice. The whole manuscript has been revised and edited by an American English speaking proof reading agency to improve the English wording. In addition, decimal points have been replaced with commas in all numbers.

3.) Third, there are few concerns/questions regarding the Materials and Methods section:

-The participants should be described more clearly, including the inclusion/exclusion criteria. 

The participants were more clearly described, including the inclusion/exclusion criteria (on page 9)

Why were all participants male and between the ages of 11 and 17? Were females attempted to be recruited?

The authors thank the reviewer for this question.

With regard to the fact that all participants are male: 

All participants in this study were men, although attempts were made to recruit women. Given the largely male-favorable sex ratio in ASD, as well as the constraint of meeting inclusion/exclusion criteria and obtaining consent, no women could be recruited in the limited time available for data collection. Indeed, time constraints related to the availability of the double mirror device did not allow us to wait for the recruitment of additional cases to obtain a more mixed sample. Because the purpose of this pilot study was primarly to see if SOD impairments could be observed in ASD cases compared to TDC and if the Double Mirror paradigm was able to show and measure such impairment, we considered that the most important methodological requirement was to control cases and control on age and gender. This has been added to the methodological section. Moreover, this limitation was specified and appears several times in the manuscript.

Why not younger individuals as discussed later in the Discussion section (page 25)?

Younger individuals were not recruited because the 10-year cutoff was chosen to ensure the acquisition of a stable body schema in typically developing subjects. This clarification has been added with the appropriate literature reference (page 9).

As mentioned in the discussion, this first study opens the possibility of comparing with a younger population in a second time to apprehend the developmental effect but this is beyond the scope of this first pilot study.

Why did one individual not receive an ADOS assessment? This isn’t then mentioned when the correlation coefficients are assessed. 

One individual did not receive an ADOS assessment as part of the standard of care on the assessment unit because his clinical condition did not allow the assessment to be done (too anxious about the camera). Moreover, there was no clinical benefit for the patient to attempt to repeat the assessment. This clarification has been added (page 9)

The TDCs were recruited by participants and staff, but where/how were they recruited?

TDCs were recruited by oral information in nearby elementary and middle schools. This was specified in the method section (page 9) 

4.) On page 15, the first point of the Figure 2a description states that both the ASD patients and TDCs start at 100% light intensity. I believe this should be 100% for TDC and 0% for ASD?

We thank the reviewer for his vigilance, indeed, it was a mistake that was corrected.

5.) The task consisted of two passages back and forth, both starting with the ASD individual at the other condition and the TDC at the self condition. Why wasn’t one pass started with the TDC at the self condition and then the TDC starting at the other condition at the second pass. In other words, I feel that possibly starting the ASD individual at the other condition each time may possibly create some sort of “bias.” I understand that they both go through each “phase” the same amount of times, but just how the process begins seems like it may impact the outcomes.

We thank the reviewer for this comment. Indeed, the order of presentation of the task was not randomized, which could lead to carry over effects. On the other hand, there was no significant effect of time on the results of the recognition task in individuals with ASD and TDC, thus reducing possible carry over and learning effects of the task. Finally, the protocol was the same as in the study published by Keromnes with regard to the direction of the passages, in order to allow an interpopulation comparison.

6.) There are also come ambiguous/subjective comments in the methods (page 16). For example, “the expected response was either “me” or “he/she.” How were they told to respond, what were the actual responses, and was it possible to not get any response? 

We thank the reviewer for this comment. Participants were asked to answer either "me" or "him/her"; it was not possible for the participant accepting to participate to the study, to give no answer or to give an answer outside of this binary choice. Cases and TDC parents were informed of this constraint before agreeing to participate to the study. Ambiguous or subjective comments in the methods were corrected.

Another example, “The type of response measured and recorded is a verbal response, which seems more relevant than a button response…” Why was this more relevant?

It seems to us more relevant because, in the same way as Keromnes did, we think that there is a real clinical interest in favoring a response based on the use of verbal language and in particular of a personal pronoun (me or him) to designate, given their importance during the developmental period, and especially the difficulties in this area in ASD individuals (the relationships between language development and successful SOD have been added on page 8). Moreover, a verbal response also seems more relevant in the field of self-recognition and self-consciousness, which are central issues of this work. This has been further explained in the manuscript (page 18).

Move the description of this reasoning from the Discussion to the Methods and change the wording to be less subjective.

The description of this reasoning has been moved from Discussion to Methods and the wording has been changed to be less subjective.

Finally, “…a systematic 10 minutes pause…”; what was systematic about it? Why 10 minutes?

A 10-minute break was given to all participant to allow attentional recovery

7.) The end of the Procedure section mentions a debrief during which participants were asked different questions. Why were these questions asked and where are the results/responses?

At the end of the task, there was time for a friendly debriefing to detect any anxiety triggered by the task. A snack with drinks and cakes was provided. Participants were asked questions including whether they found the task difficult and whether they used certain means or details to discriminate between the two faces, also whether they were used to practicing morphing sessions in daily life. This was an informal moment, which did not give rise to any quantitative or qualitative data collection. This point has been clarified in the manuscript page 19.

8.) In general, the Data and Statistical Analysis sections needs to be expanded to include more detail, and the specific analyses should be better explained. 

The sections on data and statistical analysis have been expanded to include more details, and specific analyses have been better explained in the following manner (page 20/21)

To assess potential relationships between performance in our task and autism severity,we computed correlations between M1 et M2 and the ADOS severity score. Since the data are non-gaussian, the Spearman's rank correlation coefficient was used and the associated p-value, which allows to test if the dependency is statistically significant, was computed using the algorithm AS 89 (see Best et al., 1975 ) implemented in R."

The outcome is stated to be the light intensity levels (expressed as percentage), but it’s unclear as to how these were calculated. The results in Table 2 show percentages larger than 100. Shouldn’t these values be between 0 and 100, and only in increments of 10, since the light increased 10% each step?

Since modifications of light intensity on one side of the double mirror modifies its reflective properties on both sides, we had to consider the relative light intensity that was actually perceived by participants on each side. In order to reflect this different perception for each subject, it was necessary to define a value specific to each participant at a time t of the experiment: the "relative" light intensity as shown in Fig 2b. Thus, it was necessary to find a calculation of M1 and M2 which made it possible to compare these two thresholds between themselves but also between TDC and ASD groups. The second constraint was that the variable was continuous. Similar to Keromnes et al (2018) with schizophrenia patients, this relative light intensity was obtained by combining the absolute light intensity values via the following calculation:

IA = absolute light intensity on ASD’s side 

IT = absolute light intensity on the TDC side

Relative Light Intensity for ASD participants = 100 + IA - IT 

Relative Light Intensity for TDC participants = 100 + IT – IA 

For instance, at the beginning of the experiment, when the mirror is illuminated at 100% on the TDC side and 0% on the ASD side, the relative light intensity for the TDC is 200, and that of ASD individual is 0.

9.) Finally, the Results section includes statistical tests (page 19) that were not mentioned in the data analysis section, and the p-values presented on page 19 are not included in any of the tables and are unclear as to where they came from. Also, Table 3 of the “correlation analysis” doesn’t include any of the actual correlation coefficients.

The statistical tests (page 19) have been mentioned in the Data Analysis section, and the p-values have been completed. An additional graph entitled “Figure 5: Spearman's correlation analyses of different pairs of variables (various colours) and the associated p-value (indicated by the text)” has been added to detail the correlation coefficients. 

Reviewer 2:

 I apologize for my belated review. I appreciate this opportunity to review Lavenne-Collot and colleagues’ novel and rigorous work of using a novel double mirror task to provide an ecologically valid paradigm to test self-other distinction problems in autism. To be honest, I personally learned a great deal from reading their work. Nonetheless, I have some comments, which hopefully may help improve the current form of the manuscript.

First, the authors sincerely thank the reviewer for taking the time to proofread this article so thoroughly. We were very encouraged by the reviewer's positive comments on our work. We are pleased that he saw the value of publishing this work despite its limitations.

1. I highly respect the French language and culture. Nonetheless, there are several style issues, which are not conventionally used in the English language, throughout the manuscript. Given that this article is going to published in English journal, I suggest that these issues should better be amended. Specifically,

‘i-e’ should be ‘i.e.’

‘e.g,’ should be ‘e.g.’

The Guillemets ‘« »’ should be expressed in ‘ ” ” ’ or ' ' ' ', the quotation marks which are commonly used in English

’13,8’ should be ’13.8’, and this should also applied to all number style

The authors would like to thank the reviewer for his kindness. The entire manuscript was revised and edited by an American English-speaking proofreading agency to improve the English wording.

2. ‘Embodiment’ was first mentioned in Introduction (Page 9), but was not explained. A concise explanation should be added.

We thank the reviewer for this comment. A concise explanation of the term "embodiment" has been added on page 8.

3. In Methods and Results, autism severity used in the correlation analysis with M1 or M2 was not clearly stated. I wonder whether it is social-communication in ADOS-2 or social and communication, respectively, in ADOS-2 was used in the correlation analysis.

We thank the reviewer for this very pertinent comment. In Methods and Results, the severity of autism used in the correlation analysis with M1 or M2 was shown as well as the different ADOS scores that were used in the correlation analysis.

A new figure has been added entitled: "Spearman rank correlation coefficient between the different pairs of variables (color) as well as the associated p-value (given by text)".

4. On Page 21, “In comparison with the social interactions score and then the communication score, no correlation was significant” This statement was inaccurate to me.

We thank the reviewer for this pertinent remark. A figure has been added to detail the correlation coefficients (Figure 5). Moreover, in the manuscript, the following paragraph has been added to comment the figure:

“Figure 5 summarizes the relationships among the different variables. We found a positive correlation between M1 and M2 (in blue) and a negative correlation of M1 and M2 with ADOS scores (red). The lowest p-value was found regarding the relationship between the M1 threshold during the second passage back and forth and the overall ADOS score (correlation coefficient -0.79, p= 0.059), indicating earlier face recognition when one's own image gradually appeared in the more severely affected ASD individuals. M2 levels and ADOS scores were not significantly correlated. Regarding strongly-correlated ADOS sub-scores, a slightly lower p-value was found in the relationship between the M1 level and the 'interaction' score compared to the 'communication' score. Therefore, the ADOS interaction score is most closely linked to M1.”

5. I think the biggest caveat of this study is very small sample size (n=7 in each group). Together with the limitation of male-only sample, these two caveats (male, very small sample size) should be stated very clearly in the first paragraph of Discussion and in the conclusion, in order to explicitly remind the readers that before starting discussing the results and before making the conclusive remarks, they need to be aware of these limitations, resulting in preliminary nature of this study.

We thank the reviewer for this advice. The two main caveats (male, very small sample size) have been clearly stated in the first paragraph of the discussion and in the conclusion, to emphasize to readers the limitations resulting from the preliminary nature of this study.

As a matter of fact, all participants in this study were male, although attempts were made to recruit women. Given the largely male-favorable sex ratio in ASD, as well as the constraint of meeting inclusion/exclusion criteria and obtaining consent, no women could be recruited in the limited time available for data collection. 

Indeed, time constraints related to the availability of the double mirror device did not allow us to wait for the recruitment of additional cases to obtain a more mixed sample. Because the purpose of this pilot study was primarly to see if SOD impairments could be observed in ASD cases compared to TDC and if the Double Mirror paradigm was able to show and measure such impairment, we considered that the most important methodological requirement was to control cases and control on age and gender. This has been added to the methodological section. Moreover, this limitation was specified and appears several times in the manuscript.

6. On Page 21, “Moreover, SOD impairements were correlated with ASD severity” This statement is inaccurate, as there was only a trend-level correlation, despite a large effect size. This statement should be toned down.

We thank the reviewer for this comment. The statement has been toned down to emphasize that there was only a trend-level correlation.

7. On Page 25, the authors noted that their speculation, that response inhibition impairments in ASD may in part explain the SOD problems, is inconsistent with the Review (83). Nonetheless, this cited review was mainly based on the inhibitory control task in non-social contexts. The authors may also cite https://www.sciencedirect.com/science/article/pii/S221053361830022 to argue that the evidence of inhibitory control impairments in social contexts associated with autism is actually inconclusive.

The authors thank the reviewer for this wise advice. The recommended bibliographic reference has been added.

8. I did not find which result could support the discussion on Page 27 and 28 surrounding the “disturbances in mirror self-recognition suggesting a weakened sense of self”. Could authors please clarify this part?

We thank the reviewer for his comment. This sentence has been modified to be less subjective.

9. On Page 30, “These similar findings support the idea of a continuum between autism and schizophrenia spectrum disorders (37, 114, 115).” https://jamanetwork.com/journals/jamapsychiatry/fullarticle/2773832 This reference is very relevant to this statement and could be cited here.

We thank the reviewer for this another bibliographic reference that has been added

10. On Page 30 and 31, it’s very interesting to note that one autistic participant indicated the practice and learning effect. The authors used this to support their further argument surrounding the clinical implication. I like the approach. Nonetheless, I wonder what his autistic severity was. If his autistic severity was relatively low, this can be further used to support that by training SOD using the double mirrors or other similar paradigms, people may be able to have their behaviours modified.

We thank the reviewer for this comment. Subject A7's autism severity was indeed relatively low (with low ADOS scores) and good compensatory strategies. This was specified in the manuscript to support the hypothesis that improved SOD abilities could be an important factor for compensatory skills. A more detailed assessment of participants' functioning and adaptive strategies would have been interesting to clarify this point. However, this is beyond the scope of this preliminary study but opens up possibilities for future work.

11. On Page 32 and Page 33, the authors discussed whether some other alternative theories/factors could explain the present results. I wonder whether social anxiety and ADHD symptoms could play a role in moderating being unable to extract eye’s information or information processing issues, respectively.

We thank the reviewer for this comment. The alternative hypotheses raised by the reviewer that social anxiety and ADHD symptoms could play a role in moderating being unable to extract eye’s information or information processing issues have been added (page 

12. On Page 33, “Conversely, their preferential local processing might have allowed ASD individuals to recognize their face earlier than TDCs during the other to self morphing sequence.” This statement is unclear to me. Could authors please provide a more detailed explanation for this sentence and the argument?

We thank the reviewer for this request for clarification. This paragraph has been modified to provide a more detailed and clear explanation with the following formulation:

Finally, much research contrasts local versus global processing, with a bias towards local processing in individuals with ASD (116). This could explain the differences based on morphing direction (self to other or other to self), which was found only in the participants with ASD. Indeed, in participants with ASD, both local and global processing could have been engaged to achieve the same performance, regardless of the direction of morphing. Conversely, preferential local processing might have allowed the ASD subjects to recognize their own face earlier than the TDCs during the morphing sequence from other to self (focusing on details belonging to the self, according to the SRE effect). However, this local processing could not have compensated for the lack of global processing when unfamiliar facial features appeared in the other direction.

---

## [Decision Letter · Decision Letter 1]

9 Sep 2022

Self/other distinction in adolescents with autism spectrum disorder (ASD) assessed with a double mirror paradigm

PONE-D-22-04219R1

Dear Dr. Lavenne,

We’re pleased to inform you that your manuscript has been judged scientifically suitable for publication and will be formally accepted for publication once it meets all outstanding technical requirements.

Kind regards,

Mariella Pazzaglia

Academic Editor

PLOS ONE

Additional Editor Comments (optional):

Reviewers' comments:

Reviewer's Responses to Questions

**Comments to the Author**

1. If the authors have adequately addressed your comments raised in a previous round of review and you feel that this manuscript is now acceptable for publication, you may indicate that here to bypass the “Comments to the Author” section, enter your conflict of interest statement in the “Confidential to Editor” section, and submit your "Accept" recommendation.

Reviewer #2: All comments have been addressed

2. Is the manuscript technically sound, and do the data support the conclusions?

Reviewer #2: Yes

3. Has the statistical analysis been performed appropriately and rigorously? 

Reviewer #2: Yes

4. Have the authors made all data underlying the findings in their manuscript fully available?

Reviewer #2: Yes

5. Is the manuscript presented in an intelligible fashion and written in standard English?

Reviewer #2: Yes

6. Review Comments to the Author

Reviewer #2: (No Response)

7. PLOS authors have the option to publish the peer review history of their article (what does this mean?). If published, this will include your full peer review and any attached files.

Reviewer #2: **Yes: **Hsiang-Yuan Lin

---

## [Editor Report · Acceptance letter]

15 Sep 2022

PONE-D-22-04219R1 

Self/other distinction in adolescents with autism spectrum disorder (ASD) assessed with a double mirror paradigm 

Dear Dr. Lavenne-Collot:

I'm pleased to inform you that your manuscript has been deemed suitable for publication in PLOS ONE. Congratulations! Your manuscript is now with our production department. 

Kind regards, 

on behalf of

Dr. Mariella Pazzaglia 

Academic Editor

PLOS ONE